# Current Concepts in Gastroparesis and Gastric Neuromuscular Disorders—Pathophysiology, Diagnosis, and Management

**DOI:** 10.3390/diagnostics15070935

**Published:** 2025-04-05

**Authors:** Jennifer Dimino, Braden Kuo

**Affiliations:** Center for Neurointestinal Health, Division of Gastroenterology, Massachusetts General Hospital, Harvard Medical School, Boston, MA 02114, USA; jdimino@mgh.harvard.edu

**Keywords:** gastroparesis, gastric neuromuscular disorders, functional dyspepsia, gastroparesis-like symptoms

## Abstract

Upper gastrointestinal concerns including gastroparesis-like symptoms affect a large portion of the population, and determining the culprit condition can be difficult due to largely shared symptoms, clinical course, pathophysiology, and treatment pathways. The understanding of gastric neuromuscular disorders (GNDs) is emerging as a heterogeneous group encompassing conditions from gastroparesis to functional dyspepsia with chronic nausea, early satiety, bloating, or abdominal pain, irrespective of gastric emptying. This article aims to review the current concepts in gastroparesis and GNDs including pathophysiology, diagnosis, and management. While some established standards in their diagnosis and management exist, a number of novel diagnostics are becoming available. Durable therapeutic options are notably limited for such common conditions with chronic and debilitating symptoms, and neuromodulators may play a key role in symptom control, which has been previously under-recognized and underutilized. Advances in both pharmacologic treatment targets as well as noninvasive and invasive interventions and devices show promise in improving the experience of patients with gastroparesis-like symptoms. At this time, treatment of GNDs requires comprehensive multidisciplinary care from providers to achieve successful treatment outcomes.

## 1. Background

Upper gastrointestinal (GI) concerns affect nearly half of the population in the United States (US) [1]. Symptoms of upper GI disorders may be nonspecific or share significant overlap. Gastroparesis is a chronic condition characterized by delayed gastric emptying without mechanical obstruction that presents with nausea, early satiety, bloating, or abdominal pain. However, the experience of patients with gastroparesis can vary substantially in terms of symptoms, severity, and complications. Studies have shown that gastric emptying does not reliably correlate with symptoms [2]. Furthermore, gastric emptying results have been shown to vary even in the same individual over time without corresponding changes in symptom scores, raising the question of the utility of gastric emptying to guide the diagnosis and management of upper GI symptoms [3]. Functional dyspepsia (FD) is a disorder of bothersome abdominal symptoms that can be concurrent with gastroparesis, and the predominant condition may be difficult to distinguish. The significant overlap in the symptoms and physiology of these conditions suggests that the line between gastroparesis and FD may be blurred, and shared treatment pathways may be beneficial [4]. Unless severe and persistent, delayed gastric emptying may represent a marker of underlying gastric neuromuscular dysfunction rather than the cause of symptoms. The contribution of FD needs to be addressed for successful symptomatic treatment of patients with gastroparesis. In fact, patients with a constellation of upper GI symptoms including chronic nausea, early satiety, bloating, and abdominal pain may be better considered as one group of gastric neuromuscular disorders (GNDs) irrespective of gastric emptying results. Despite growing awareness, GNDs remain a challenging entity to manage due to their heterogeneous etiology and variable clinical presentation.

## 2. Epidemiology

The prevalence of gastroparesis is difficult to ascertain due to diagnostic challenges including limited diagnostic data and misdiagnosis, but US study estimates suggest a range from 6.3 to 17.2 cases per 100,000 person-years, with a higher prevalence in women [5]. The overall age-adjusted incidence of gastroparesis in the US population was 2.4 per 100,000 person-years for men, 9.8 per 100,000 person-years for women, and 6.3 per 100,000 person-years for the general cohort. Gastroparesis occurs in people of all backgrounds in a large US population-based study, though testing to confirm the diagnosis was only documented in the electronic medical record in approximately 21.5% of charts carrying a gastroparesis label [6]. Women and Caucasians were noted to have the highest prevalence of gastroparesis, and multiple studies support the female predominance of gastroparesis with as high as 80% of cases affecting female individuals [5,7,8]. Diabetes mellitus accounts for a significant proportion of cases, particularly type 1 diabetes, followed by idiopathic and post-surgical causes. Studies report that up to 12% of patients with diabetes experience symptoms consistent with gastroparesis, but not all undergo confirmatory testing [9]. The overall prevalence of gastroparesis in the US population is estimated at 0.16%, with notable increases in prevalence associated with type 1 diabetes mellitus (4.59%) and type 2 diabetes mellitus (1.31%) [6]. However, given the large number of patients with type 2 diabetes in the US, the number of patients in that group may range from 3- to 9-fold greater than the type 1 diabetes group [6,10].

## 3. Symptoms

Gastroparesis typically represents a chronic condition with symptoms that are difficult to control. A majority of patients (72%) did not report clinically meaningful symptomatic improvement over 48 weeks to 4 years of follow-up [11]. Up to 15% of patients reported an acute onset of symptoms following a prodrome of respiratory symptoms, gastroenteritis, or food poisoning [12]. The symptoms of gastroparesis are diverse and overlap with other gastrointestinal disorders such as functional dyspepsia. The most commonly reported symptom is nausea (greater than 95%), followed by upper abdominal pain (90%), vomiting (80% diabetic gastroparesis, 57% idiopathic gastroparesis), bloating (three-quarters), and early satiation (two-thirds). Symptom severity does not correlate with the severity of gastric emptying delay, corroborating the concept of FD as a primary contributor to symptomatology. Quality of life is significantly affected in these patients, with more than half having clinically significant anxiety scores, and almost half having moderate-to-severe depression scores [13]. Despite this, only 40% of patients with significant depression scores were prescribed an antidepressant medication, identifying a gap in care for many patients which should be addressed. Uncontrolled symptoms and flares also lead to hospitalization and increased healthcare usage and costs [14]. However, it is worth noting that while the length of hospital stays for gastroparesis decreased by 20% between 1997 and 2013, the mean hospital charges increased significantly by 159%, underlying the observed increase in healthcare costs.

## 4. Etiology

Gastroparesis can be classified into idiopathic (11.3–49.4%), diabetic (25.3–78.1%), post-surgical (15.0%), and less commonly, medication-induced or viral subtypes [8,10]. The most common type of gastroparesis is idiopathic, with no cause identified in approximately one-third to two-thirds of cases in the community and referral centers, respectively, though, in many cases, the cause is suspected or partially understood [15,16]. Commonly known causes of gastroparesis include neuropathic conditions such as diabetes (29%), post-vagotomy (13%), Parkinson’s disease (7.5%), and collagen vascular disorders (4.8%) such as scleroderma. Other conditions that cause gastroparesis include neurologic disorders, myopathic disorders, post-viral syndromes, connective tissue diseases, metabolic or endocrine disorders such as hypothyroidism, critical illness, and eating disorders, and medications, particularly opioids and anticholinergic agents [4]. Post-viral gastroparesis tends to improve within a year, but a small subset of patients infected with Epstein–Barr virus (EBV), cytomegalovirus (CMV), and varicella-zoster virus (VZV) can develop autonomic dysfunction which can persist [17]. Iatrogenic or medication-induced gastroparesis is seen most commonly with opioids, as well as some metabolic agents, neuromodulators, antibiotics, anticonvulsants, and antiarrhythmics. It is generally reversible if the medication can be held, and it is often tolerable in the case of glucagon-like peptide-1 (GLP-1) agonists.

## 5. Pathophysiology

While the exact cause of gastroparesis is unknown, evidence supports a few theories behind the development and pathophysiology of the disease. Key functional abnormalities in gastroparesis involve disruptions in gastric motility and coordination, which are critical for the efficient processing and emptying of stomach contents. A broad set of observed abnormalities include alterations in gastric emptying, gastric accommodation, gastric coordination, gastric rhythm, autonomic function, nongastric motility and feedback, and visceral hypersensitivity. These abnormalities are summarized in Figure 1.

### 5.1. Gastric Emptying

Gastric emptying delays have historically been the hallmark of gastroparesis. Delayed gastric emptying often results from impaired contractions of the antrum and inadequate relaxation of the pyloric sphincter. Weak antral pump function, characterized by reduced contractile strength of the stomach’s antral region, impairs the grinding and propulsion of food particles towards the duodenum. Increased pyloric tone and impaired relaxation contribute significantly to gastric outlet obstruction [18]. Pyloric dysfunction is often associated with hypertrophy or fibrosis, which limits its compliance and obstructs food passage. These disruptions can lead to stasis of food in the stomach, causing symptoms such as bloating, nausea, and vomiting. However, newer evidence suggests a questionable or nonexistent correlation between gastric emptying delays and symptom severity. In fact, individuals were noted to transition between normal and delayed gastric emptying in longitudinal studies, without a corresponding change in symptom scores [3]. This begs the question of whether gastric emptying is relevant in the symptomatology of patients with gastroparesis. However, in diabetic patients, delayed gastric emptying does corelate with extragastric neuropathies, and may be a marker of the overall disease state [19]. While severe and persistent gastric emptying delays may contribute to symptomatology, mild and/or intermittent delays may serve clinically as an indicator of underlying gastric neuromuscular dysfunction as opposed to the cause of symptoms. Therefore, gastric emptying may be reassessed over time, most reasonably if there are significant changes in symptoms, generally no sooner than at 6 months, and more judiciously following multiple years.

### 5.2. Gastric Accommodation

Emerging evidence also points to abnormalities in gastric accommodation—the stomach’s ability to expand and store food after meals—which may contribute to early satiety and postprandial fullness. This process is mediated by both Vagal and intrinsic nitrergic neuronal activity [20]. Functional abnormalities have been shown via diminished volume tolerated on water load satiety testing (WLST) as well as low intragastric meal distribution on gastric scintigraphy studies in patients with gastroparesis [21,22]. It is worth noting that routine gastric scintigraphy may therefore provide clinicians with additional information outside of standard gastric emptying information. As gastric accommodation is largely implicated in the symptomatology of FD, particularly postprandial distress syndrome, this is yet another piece of the puzzle in the shared pathophysiology of GNDs [23].

### 5.3. Gastric Coordination

Antroduodenal discoordination, where the stomach and small intestine fail to synchronize their movements, is another critical abnormality. This lack of coordination exacerbates gastric stasis and contributes to the backflow of contents, further delaying digestion and causing discomfort. The migrating motor complex is a recurring cyclic motility pattern with four distinct phases that occurs in the stomach and small intestine during fasting but is interrupted by feeding [24]. MMC is mediated largely though not entirely through vagal nerve pathways; vagotomy extinguishes gastric motor activity but leaves periodic small intestinal activity intact. An antroduodenal manometry study demonstrated antral hypomotility in over three-quarters of patients with gastroparesis (90% of diabetic gastroparesis, 75.2% of idiopathic gastroparesis, 76.5% of post-surgical gastroparesis) [25]. The study also revealed a prolonged time from food ingestion to the return of MMC, as well as fewer and shorter durations of the strongest phase III contractions in both idiopathic and diabetic gastroparesis compared to post-surgical gastroparesis. However, the number of phase III contractions and MMCs did not correlate with clinical severity or gastric emptying. These data suggest that disturbances or delays in antroduodenal contractions may play a larger role in the symptoms of gastroparesis than previously appreciated.

### 5.4. Gastric Rhythm

Disruptions in gastric pacemaker activity and gastric arrhythmia are important aspects in the pathophysiology of GNDs. The interstitial cells of cajal (ICCs) serve as the gastrointestinal pacemaker generating and propagating slow waves distally from the greater curve of the stomach body typically at three cycles per minute [26]. Abnormalities in slow-wave initiation and conduction result in gastric dysrhythmias, which impair the stomach’s ability to generate coordinated contractions. Gastric rhythm, as measured by electrogastrography (EGG), showed differences in electrical activity in gastroparesis, including increases in brady- or tachygastria activity or a failure to increase gastric myoelectrical activity after ingestion of test meals [27]. These rhythm disturbances reflect abnormal electrical activity within the gastric musculature and are associated with disrupted gastric contractions and delayed emptying. Interestingly, 84% of patients with diabetic gastroparesis had gastric dysrhythmias, and 16% had normal three-cycles-per-minute gastric myoelectrical activity, findings that were reproducible at baseline and 48 weeks despite continuous glucose monitoring and continuous subcutaneous insulin infusion, suggesting varying phenotypes which may be relevant.

### 5.5. Autonomic Function

The impairment of gastric emptying can be caused by abnormalities in the neural control mechanisms that regulate gastric motility, primarily through the Vagus nerve. Appropriate gastric contractile activity is regulated by the complex interplay between the enteric nervous system, ICCs, smooth muscle, and vagal inputs [28]. Known vagovagal reflexes include esophagogastric receptive relaxation, gastrogastric accommodation, and duodenogastric duodenal brake; these reflexes are mediated by the activation of mechanoceptors and/or chemoreceptors within the walls of the GI tract, with subsequent feedback regulating gastric compliance, motility, and emptying. Autonomic dysfunction including sympathetic withdrawal and parasympathetic dysfunction is commonly seen in patients with gastroparesis and chronic unexplained nausea and vomiting [29]. Patients with diabetes were found to have greater global autonomic dysfunction. Patients with severe symptoms were noted to have more parasympathetic dysfunction in relation to patients with mild–moderate symptoms that demonstrated more sympathetic hypofunction.

### 5.6. Extragastric Motility and Feedback Mechanisms

Extragastric dysmotility was frequently seen in patients with gastroparesis symptoms by wireless motility capsule (WMC) testing with around a third having delayed colonic transit, approximately 15% having delayed small bowel transit, and 25% having a combination of small bowel and colon transit delays [30]. Gastroparesis may therefore represent just one component of a more generalized enteric nervous system dysfunction. Patients with delayed colonic transit had higher constipation scores, and patients with delayed small bowel or colon transit had lower diarrhea scores. As these findings may impact and contribute to patients’ symptoms, addressing extragastric dysmotility may alleviate some of their GI concerns. Furthermore, altered feedback mechanisms between the small intestine and stomach, such as excessive duodenal sensitivity to acids or fats, may worsen dyspeptic symptoms and delay gastric emptying [31]. Additionally, undigested nutrients in the ileum activate the “ileal brake”, a complex mechanism influencing the digestive process and ingestive behavior [32]. In a process involving vagal and enteric neurons, as well as central and spinal pathways, enteroendocrine cells in the intestine release glucagon-like peptide 1 (GLP-1) in response to nutrients, inhibiting gastric emptying and feeding [33].

### 5.7. Visceral Hypersensitivity

Other functional impairments include heightened visceral hypersensitivity, where patients experience exaggerated pain and discomfort in response to normal gastric distension. In retrospective gastric accommodation testing performed using CT-scan gastric volumetry to assess the distension of different regions of the stomach, patients with gastroparesis reported a significantly higher self-assessment of pain related to gastric distension compared to patients with gastroesophageal reflux disease [34]. This hypersensitivity may be mediated by altered signaling pathways between the gut and central nervous system including the Vagus nerve, and can compound the severity of symptoms.

## 6. Cellular and Molecular Pathology

The pathophysiology of gastroparesis and GNDs is multifactorial, involving complex interactions between gastric motility, neuromuscular function, hormonal regulation, and immune responses. Further studies into the cause of these abnormalities are needed, as current studies have been performed primarily in patients with refractory symptoms. The most commonly noted abnormalities were loss of expression of neuronal nitric oxide synthase (nNOS) and loss of ICCs. Studies are currently underway to evaluate these changes on a histologic and molecular level.

### 6.1. Neuropathy

#### 6.1.1. Intrinsic Neural Deficits

Neurologic disorders such as Parkinson’s disease, amyloidosis, dysautonomia, and diabetic neuropathy are theorized to affect both intrinsic and extrinsic neural pathways. Nitric oxide, a major inhibitory nonadrenergic, noncholinergic neurotransmitter of the GI tract that causes smooth muscle relaxation, is synthesized by the activation of neuronal NO synthase (nNOS) in the myenteric plexus, and the reduced expression of nNOS with impaired local NO production is implicated in GI motility disorders [35]. NO mediates gut function via multiple pathways, including the regulation of sphincter muscle tone (lower esophagus, pylorus, sphincter of Oddi, anus), gastric accommodation of the fundus, gastric emptying, peristaltic reflex of the intestine, and colon transit. Light microscopy has revealed neuronal damage including loss of nNOS neurons in 20% of diabetic gastroparesis patients and 40% of idiopathic gastroparesis patients [36]. Electron microscopy revealed additional changes in patients with idiopathic gastroparesis, including damaged cell bodies, mitochondria, and nerve endings [37], whereas in diabetic gastroparesis, neuronal injury was less prominent and primarily affected the nerve endings, sparing the nerve bodies. Quantitative immunohistochemistry analyses have not shown a difference in neuronal subtypes (neuronal nitric oxide synthase/nNOS, substance P/SP, vasoactive intestinal peptide/VIP, tyrosine hydroxylase/TH) between idiopathic gastroparesis, diabetic gastroparesis, and control groups.

#### 6.1.2. Vagal Neuropathy

Autonomic or vagal neuropathy is most prominent in cases of post-surgical gastroparesis, and is generally irreversible in the setting of complete vagal nerve injury or vagotomy, as seen in fundoplication or lung transplant. In diabetic gastroparesis, autonomic neuropathy disrupts vagal nerve function, leading to impaired coordination between gastric contractions and pyloric relaxation. Hyperglycemia, a hallmark of diabetic gastroparesis, exacerbates motility dysfunction by impairing vagal activity and reducing antral contractions [38]. An open-label pilot study on patients with idiopathic gastroparesis demonstrated that noninvasive vagal nerve stimulators can modulate gastroparesis symptoms and gastric emptying, further corroborating the contribution of vagal neuropathy to disease [39]. Dysautonomia seen in POTS, Parkinson’s disease, and some post-viral syndromes is also associated with gastroparesis. Post-viral gastroparesis conditions are thought to originate from direct damage to systemic neurons, and have poorer prognosis when associated with acute autonomic dysfunction such as cases of norovirus, EBV, and CMV [40].

### 6.2. Interstitial Cells of Cajal (ICCs)

The ICCs are the pacemaker of the stomach and gastrointestinal tract, generating slow-wave impulses toward the pylorus to induce gastric emptying and downstream motility. Full-thickness gastric biopsies have shown depletion of ICCs and muscle fibrosis in patients with both idiopathic and diabetic gastroparesis [41]. Significant (>50%) ICC loss was also found in the gastric antrum in approximately 60% of patients in both groups compared to controls [42]. Loss of these intrinsic neurons has been shown to occur via inflammatory macrophage-mediated pathways as well as degenerative reactive oxygen species pathways. An ion channel Ano1 mediates Ca^2+^-activated Cl^−^ pathways in ICCs, and a novel variant Ano1 Δ1,2,3 (5′) (whose expression generates a decreased density of Ca^2+^-activated Cl^−^ channels with slower kinetics compared to standard Ano1) was found at several-fold higher levels in the gastric muscle of patients with diabetic and idiopathic gastroparesis compared with normal controls [43]. However, the trigger, cause, and full sequelae of these changes remain unclear.

### 6.3. Smooth Muscle Dysfunction

Myopathic conditions including scleroderma and amyloid (which can also affect the nerves) can directly impact and impair motility. Full-thickness gastric biopsies in patients with gastroparesis have shown muscle fibrosis in addition to depletion of ICCs [41]. Both diabetic and idiopathic gastroparesis groups had a marked increase in collagen fibrils in the connective tissue stroma, which raises the question of the role of gastric and pyloric myopathy as a primary abnormality of gastroparesis [32]. In a study of 40 patients with diabetic and idiopathic gastroparesis, all patients had a majority of normal-appearing smooth muscle cells, with only a few (2–3 cells/section) with morphologic alterations such as lipofuscin and lamellar bodies or swollen mitochondria [36]. Patients with diabetic gastroparesis were distinctly noted to have marked thickening of the basal lamina in smooth muscle cells. Patients with idiopathic gastroparesis were found to have altered smooth muscle contractile protein expression and loss of platelet-derived growth factor receptor α^+^ cells without change in ICCs [44]. More studies are needed to understand the role of myopathy in gastroparesis.

### 6.4. Inflammatory and Immune Mechanisms

Inflammatory and immune-mediated mechanisms are increasingly recognized as contributing factors in gastroparesis. In idiopathic cases, inflammatory infiltration of the gastric muscularis propria has been observed, with associated loss of ICCs and neuronal elements [45]. Immunohistochemical stains have suggested an increased number of macrophages and other phagocytic cells in the gastric body of patients with gastroparesis [36]. In the muscular and myenteric plexus layers of the gastric antrum of patients with gastroparesis, anti-inflammatory CD206+ M2 macrophage cells were found to be significantly reduced, which also correlated with ICC counts [42]. The gene HMOX1 encodes the heme oxygenase 1 (HO1) enzyme, expressed in CD206+ macrophages, whose loss leads to loss of ICC in animal models [46]. The mechanisms underlying this loss may therefore have causal implications in the pathogenesis of disease. Deep transcriptome analysis of the gastric body and proteomic analysis of the gastric antrum revealed molecular changes in pathways involving macrophages, fibroblasts, and endothelial cells; some molecules related to inflammation, such as prostaglandins and complement pathway proteins, appeared to be related to gastric emptying delay [47].

Neuroimmune processes targeting neural and interstitial cells are also implicated, particularly in post-infectious gastroparesis. Post-viral gastroparesis is caused by viral triggering of neuroimmune pathology and/or autoimmunity due to immune-mediated damage of the enteric nervous system [48]. Autoimmune GI dysmotility is an underappreciated autoimmune dysautonomia with findings of abnormal GI motility and symptoms of nausea, vomiting, abdominal pain, constipation, and diarrhea. Approximately 50–70% will have serologic evidence of neural autoantibodies, such as acetylcholine receptor (AChR) and glutamic acid decarboxylase 65 (GAD65) [49,50]. Some patients with autoimmune GI dysmotility will have overlapping autoimmune conditions including type I diabetes, Grave’s disease, Celiac disease, and lupus, or, in rare cases, a paraneoplastic etiology.

### 6.5. Hormonal and Metabolic Factors

Acute hyperglycemia is known to inhibit gastric emptying, and chronic hyperglycemia is thought to have more insidious effects mediated by systemic (intrinsic and extrinsic) nerve damage to both motor and sensory components, perhaps explaining the increased risk of gastroparesis in both type 1 and type 2 diabetes mellitus. Hormonal dysregulation, including altered signaling of motilin and ghrelin, may play a role in gastroparesis pathophysiology. These hormones are critical for stimulating gastric motility and coordinating meal-related gastric activity. Altered ghrelin levels, often observed in FD patients, may impair gastric emptying and contribute to symptoms such as early satiety [51]. Furthermore, trials of administration of ghrelin or its receptor agonist in human clinical trials and animal studies show favorable results improving the symptoms of gastroparesis, supporting the theory of its dysregulation in the pathophysiology of GNDs [52].

### 6.6. Mitochondrial Dysfunction

Emerging research highlights the role of mitochondrial damage in the pathogenesis of gastroparesis. Smooth muscle mitochondria were noted to be swollen in full-thickness gastric biopsies of patients with diabetic and idiopathic gastroparesis [36]. Hyperglycemia-induced oxidative damage may impair cellular structures, including ICCs and enteric neurons, leading to progressive motility dysfunction [53]. Impaired energy production in gastric smooth muscle and neurons contributes to their dysfunction. This multifactorial etiology highlights the complexity of gastroparesis and underscores the need for targeted research to elucidate its underlying mechanisms.

### 6.7. Genetics

Current studies are evaluating if a genetic predisposition or cause for gastroparesis exists, and have found implicated genes and pathways related to immune and sensory–motor dysregulation, though further studies are needed to fully understand their significance or contribution [54]. Downregulation of HMOX1 gene expression plays a causal role in diabetic gastroparesis in animal models, and additional studies have demonstrated that polyGT alleles of HMOX1 were longer in patients with gastroparesis, which was most pronounced in patients with type 2 diabetic gastroparesis [55]. This suggests that allele length may correlate inversely with HMOX1 expression, and the presence of long polyGT repeat alleles was associated with more severe nausea symptoms. An allele of PXDNL (Peroxidasin-like protein) was associated with increased abdominal pain severity scores, and gastric muscularis expression of PXDNL was also positively correlated with abdominal pain in patients with gastroparesis. One protein, Dickkopf WNT Signaling Pathway Inhibitor 1, had decreased expression in patients with diabetic gastroparesis compared to controls.

### 6.8. Microbiome and Dysbiosis

Infections including viral, bacterial, and protozoal etiologies have been implicated in the development of gastroparesis. The microbiome, while not directly implicated in gastroparesis, may be related to FD, as microbiota dysbiosis plays a role in the development and progression of the disease [53]. However, no obvious microbiome differences or targets relevant to gastroparesis have been found.

### 6.9. Shared Pathways with Functional Dyspepsia (FD)

FD is thought to be caused by a combination of motor and sensory disturbances, similar to gastroparesis. An increased number of duodenal eosinophils and mast cells seen on biopsies may suggest an immunoinflammatory mediated pathway in FD as well [56]. The “leaky gut” theory proposes that the impaired intestinal mucosal barrier integrity seen in FD may be related to microbiome alterations [57]. Increased intestinal permeability is associated with mucosal immune activation and FD symptoms; however, the role of the microbiome in this process is not yet fully understood [58]. The shared neuropathology underlying gastroparesis and FD, including loss of ICC and CD206+ macrophages in full-thickness stomach biopsies, emphasizes the lack of distinction between them [3]. Additionally, central and peripheral nervous system dysfunction contributes to both conditions. Central sensitization may enhance visceral hypersensitivity, amplifying pain and discomfort associated with gastric distension in both disorders. Abnormalities in brain–gut communication pathways, including altered neurotransmitter signaling, may further exacerbate symptoms. The symptoms and clinical course were also indistinguishable between groups with and without delayed gastric emptying on long-term follow-up, and gastric emptying was found to transition between normal and delayed in the same patient over time without a corresponding change in symptoms. These commonalities underscore the shared neuropathology and symptomatology of the two disorders, which corroborates the notion of FD and gastroparesis or gastroparesis-like symptoms as part of a larger entity of GNDs without attention to gastric emptying status.

## 7. Diagnosis

The most common symptoms of gastroparesis include postprandial nausea, vomiting, bloating, early satiety, and upper abdominal discomfort. The presence of any of these symptoms may prompt evaluation of gastric emptying after structural and organic abnormalities are ruled out with endoscopic evaluation. Gastric emptying must be objectively delayed to give patients a diagnosis of gastroparesis. Surprisingly, in a large US population-based study, 78.5% of patients labeled with gastroparesis did not have a documented gastric emptying study [6]. The current and emerging diagnostics are summarized in Table 1.

### 7.1. Nuclear Gastric Scintigraphy

Nuclear gastric scintigraphy is the gold standard and the most widely used diagnostic modality to assess gastric emptying. Solid-phase testing is preferred and performed at most centers with ingestion of a radioisotope 99mTc sulfur colloid-labeled egg sandwich with standard imaging at 0, 1, 2, and 4 h post-prandially [59]. This method is well validated; however, some patients may struggle to ingest the entire solid meal, which is crucial for accurate results. Some variations of the standard meal exist; however, the significance of the results is less clear. Furthermore, liquid-based testing is not recommended, as gastric emptying rates of liquids are often preserved and can produce falsely normal results. Additionally, some centers do not perform the test for the full 4 h and instead calculate T1/2, which can provide variable results. Since hyperglycemia can affect gastric emptying, a fasting blood glucose less than 180 mg/dL is recommended prior to initiating a gastric emptying evaluation [60]. Medications that affect GI motility, such as metoclopramide, erythromycin, opioids, GLP1 agonists, and anticholinergic, should be withheld at least 48 to 72 h before the exam [61]. While limitations exist, nuclear gastric scintigraphy remains the gold standard in assessing gastric emptying, and the accuracy of newer testing modalities is validated against it.

The severity of the gastric emptying delay can be characterized based on the gastric retention on scintigraphy at four hours. A 10–15% gastric retention at 4 h is considered mild, while 15–35% gastric retention is moderate, and >35% gastric retention signifies severe disease. While patients with mild gastric emptying delays may be adequately managed with dietary modifications, more severe gastric emptying delays may signify the need for more advanced therapies, although treatment should generally be approached in a stepwise fashion.

### 7.2. Stable Isotope Gastric Emptying Breath Testing (GEBT)

Stable isotope breath testing is validated as an alternative to nuclear scintigraphy. It is easily performed with reliable results and no radiation exposure, meaning it is safe for children and pregnant or breastfeeding people, but results may be affected by physical activity, malabsorption, and advanced lung, liver, or cardiac disease [62]. Typically, a pre-meal baseline breath sample is collected, then a 13C labeled spirulina or octanoic acid test meal is ingested, and additional breath samples are taken at 30 min intervals over a 4 h period. In the absence of delayed gastric emptying, the labeled meal is quickly digested, absorbed, and metabolized to produce exhaled 13CO_2_, which is detected in the breath. In the setting of delayed gastric emptying, a delay in the accumulation of 13CO_2_ corresponding with the delay in gastric emptying is seen.

### 7.3. Wireless Capsule Devices

Wireless motility capsules (WMCs) have been shown to assess gastric transit time and additional gastrointestinal information by detecting changes in pH, temperature, and pressure; however, the Food and Drug Administration (FDA)-approved SmartPill was discontinued by Medtronic in 2023 [63]. As with scintigraphy, gastric emptying by WMC did not correlate with most upper GI symptoms except early satiety [64]. A novel, ingestible telemetric gas-sensing capsule (Atmo Biosciences) that measures the concentration of hydrogen and carbon dioxide gas species and luminal anaerobicity is emerging as an alternative for measuring gastric and intestinal transit times [65]. Recent evidence demonstrated that findings from the gas-sensing capsule regarding gastric emptying and colonic transit correlated with WMC findings in patients with FD and/or functional constipation, suggesting its suitability as an alternative to WMC in assessing regional and whole-gut transit times in patients with suspected motility disorders [66]. While the capsule is easily ingestible, further assessment of small bowel or colon transit may be limited in the same exam for patients with severe delays in gastric emptying. As with all pill devices, providers must avoid their use in patients with significant strictures or stenoses given the risk of a retained device. Although it has not yet been FDA-approved, comparative study data show that the Atmo capsule correlates well with existing technology in assessing gastric and colon motility and is clinically useful in evaluating gastrointestinal transit [67].

### 7.4. Electrogastrography

Electrogastrography (EGG) involves electrodes placed on the torso to noninvasively detect gastric myoelectrical slow-wave activity and demonstrate gastric emptying. EGG tracings have shown sinusoidal waveforms with a predominant frequency of three cycles per minute, which increases with the ingestion of meals in healthy volunteers, and have demonstrated rhythm disturbances or blunting of meal-evoked EGG signal amplitude increases that correlate with delayed gastric emptying in patients with nausea, vomiting, or dyspeptic symptoms [68]. The Gastric Alimetry device performs noninvasive body surface gastric mapping using dense electrode arrays to provide a high-resolution evaluation of gastric electrical activity, coupled with validated symptom profiling via an app, in order to assess gastroduodenal function over a standard 4.5 h test [69,70]. These data have been analyzed to identify characteristic phenotypes which may be clinically relevant, with two main symptom profile categories [71]. Gastric activity-related (sensorimotor, postgastric, and activity-relieved) profiles are thought to be mediated by gastroduodenal mechanisms such as disorders of gastric accommodation, hypersensitivity, or small intestinal pathology. Brain–gut dysregulation or vagal pathologies are frequently implicated in symptom profiles that are gastric activity-independent (continuous, meal-relieved, and meal-induced). Gastric Alimetry outperformed standard gastric emptying scintigraphy, the gastric emptying breath test, and electrogastrgraphy in terms of diagnostic profiling capability, lower intra-individual variation, and reproducibility [72]. Additionally, Gastric Alimetry was useful in streamlining clinical care and reducing healthcare utilization and costs in nausea and vomiting syndromes [73]. The GutTracker by G-Tech is another device that detects and analyzes gastrointestinal electrical activity; it shows promise in diagnosing delays in gastric emptying and other motility disturbances, but it is pending further validation and broad availability [74]. These devices also provide data regarding small bowel and colon activity, which can be beneficial as many patients with gastroparesis experience dysmotility in other areas of the GI tract. The Gastric Alimetry and GutTracker devices are FDA-cleared to evaluate gastric emptying and gastrointestinal disorders. These noninvasive patch devices may provide a simple alternative to traditional gastric emptying studies once more broadly covered by health insurance.

### 7.5. Antroduodenal Manometry

Antroduodenal manometry uses standard manometric methods to evaluate stomach and duodenal motility but may not be available at all centers. The manometry catheter is usually placed trans-nasally with fluoroscopic guidance to avoid endoscopic air insufflation [75]. Antroduodenal manometry does not reliably show differences in identifying patients with gastroparesis, but pylorospasm has been seen in patients with diabetic gastroparesis and recurrent nausea and vomiting [76]. However, in patients with known gastric emptying delays, antroduodenal manometry studies have shown characteristic patterns among gastroparesis etiologies including idiopathic, diabetic, and post-surgical [25]. The fed period (number of minutes between meal ingestion and first phase III contraction) was significantly longer in idiopathic (*p* < 0.01) and diabetic gastroparesis (*p* < 0.05) patients compared with post-surgical gastroparesis patients. Additionally, a significantly lower number and duration of phase III contractions and number of MMCs in idiopathic and diabetic patients compared to post-surgical patients were found, including the absence of MMCs during the 6 h recording period. Its clinical utility may be most notable for small intestinal pseudo-obstruction, which is otherwise a challenging diagnosis to make [77]. However, studies have shown positive antroduodenal manometry outcomes including a new diagnosis made in 14.9% of patients, a new therapy initiated in 12.6% of patients, and referral to a specialist in 8% of patients [78].

### 7.6. Pyloric Endoluminal Functional Imaging Probe (EndoFLIP^®^)

The functional luminal imaging probe (FLIP) is an FDA-approved measurement tool that utilizes high-resolution impedance planimetry to assess biomechanical properties of GI sphincters [79]. EndoFLIP has been well validated in the assessment of disorders of esophagogastric junction, but normative pyloric data have been limited [80]. Measurements are typically taken during endoscopy with limited or no opioid sedation, as opioids may theoretically affect pyloric parameters, though some studies have been performed with unsedated transoral probe placement [81]. While pyloric compliance and cross sectional area (CSA) are reduced in patients with gastroparesis, and decreased CSA and distensibility index (DI) correlate with nausea and vomiting, studies have failed to show concurrence between EndoFLIP indices and gastric emptying [82]. Pyloric EndoFLIP may currently be more useful in patient selection for gastric per oral pyloromyotomy (G-POP, which will be discussed later) in that patients with low pyloric distensibility experience a more significant benefit from G-POP [83].

### 7.7. Liquid Satiety Testing

The functional assessment of gastric accommodation and sensation is challenging but may provide additional insights into the symptomatology of patients with gastroparesis-like symptoms. While gastric scintigraphy can provide some information on gastric accommodation, there is usually no sensory reporting included. Liquid satiety testing is inexpensive, well tolerated, easy to perform, and provides information on both gastric accommodation and sensation [84]. The water load test and nutrient drink test are both forms of liquid satiety testing; either water or a nutrient drink such as Ensure™ (Abbott Laboratories, Abbott Park, IL, USA) is administered serially, and data are collected regarding fullness, maximum tolerated volume, and symptoms [85]. These tests are typically performed in research settings but may be helpful in understanding and managing patients with gastroparesis-like symptoms regardless of their gastric emptying status.

### 7.8. Gastric MRI

Gastric MRI is a noninvasive imaging technique that enables the dynamic assessment of stomach function, including motility, emptying, and even gut–brain interactions. The test utilizes a four-dimensional (4D), volumetric cine imaging, free-breathing MRI sequence with gadolinium-free contrast enhancement to assess gastric response to a food stimulus. Recent advancements in 4D cine MRI have allowed researchers to visualize dynamic differences in patients with FD, such as slower peristaltic propagation velocity and altered connectivity between the brain stem and cortical regions, suggesting disruptions in gut–brain communication. However, the test is only available in research centers and has not been validated for the evaluation of gastric emptying. This technology provides a promising new approach to comprehensively studying GNDs and may aid in developing targeted treatments by elucidating the neural and mechanical aspects of gut dysfunction.

### 7.9. Overlap with FD

Other conditions are often considered during the evaluation of gastroparesis and may coexist with gastroparesis. FD is characterized by the Rome Foundation as a condition of bothersome symptoms including postprandial fullness, early satiation, epigastric pain, or epigastric burning in the absence of structural disease [86]. FD can be further defined depending on the predominant symptoms as either postprandial distress syndrome (PDS) with postprandial fullness and/or early satiation occurring at least 3 days a week, or epigastric pain syndrome (EPS) with epigastric pain and/or epigastric burning occurring at least 1 day a week. A majority of patients with gastroparesis (87%) meet the criteria for FD; among them, 95% qualified for PDS, and 68% fulfilled the EPS criteria [3]. In prospective studies, there was little difference between FD and gastroparesis patients, particularly with respect to symptom severity and quality of life, clinical course with limited improvement over a 48-week follow-up, and gastric neuropathology with loss of ICCs and CD206+ macrophages. Functional dyspepsia can coexist with gastroparesis and may go unrecognized and untreated; clinicians must have a high index of suspicion for FD when patients present with upper GI concerns. Patients with a variety of gastroparesis-like symptoms such as chronic nausea, early satiety, bloating, and abdominal pain can be considered as a group of GNDs, without regard to gastric emptying status.

### 7.10. Differential Diagnoses and Comorbid Considerations

Obtaining a comprehensive history and physical is paramount in the diagnosis of GNDs and helps guide the evaluation, which usually includes endoscopy and imaging. Age- and gender-appropriate investigations must also be undertaken. For example, providers may recommend pelvic ultrasound for a woman of any age with early satiety, bloating, and abdominal pain to exclude an ovarian lesion in the appropriate clinical setting. After the exclusion of structural and organic causes of symptoms, additional disorders of gut–brain interaction can be considered. Chronic nausea vomiting syndrome (CNVS) is defined by Rome IV Criteria as bothersome nausea and/or vomiting occurring at least once per week for the last 3 months with initial symptoms occurring at least 6 months prior to diagnosis, in the absence of organic causes [86]. The shared symptomatology with gastroparesis suggests that underlying mechanisms may be at play. Cyclic vomiting syndrome (CVS) is characterized by acute episodes of intractable vomiting with intervening periods of normalcy and is associated with a personal or family history of migraine. It is notable that 36% of patients with gastroparesis have overlapping diagnoses of CVS. Cannabis hyperemesis syndrome (CHS) is similar in presentation but preceded by prolonged (often daily, heavy) cannabis use and resolves with sustained cannabis cessation. CHS is well known for alleviation of symptoms with a hot shower, but this can be seen in CVS as well and does not reliably distinguish the two entities. Rumination syndrome involves persistent regurgitation of recently ingested food to the mouth followed by spitting or remastication and swallowing, in the absence of retching. The diagnosis is primarily made by history, but the diagnosis can be confirmed through characteristic esophageal manometry findings. Lastly, patients with gastroparesis-like symptoms frequently have symptoms of food intake disorders such as avoidant restricted food intake disorder (ARFID) [87]. It is important to evaluate for these conditions and refer patients for treatment if such symptoms are present.

## 8. Management

Symptom scores should be tracked as an objective marker of symptoms and outcomes. Treatment should focus on controlling the predominant symptom of nausea, vomiting, early satiation, or abdominal pain. The Gastroparesis Cardinal Symptom Index (GCSI) and GCSI daily diary (DD) are useful tools to evaluate and track symptoms of nausea, vomiting, early satiety, bloating, and distension over time. The Nepean Dyspepsia Index is another validated tool that can be used to track symptoms, which measures the frequency, severity, and bothersomeness of 15 upper GI symptoms on Likert scales [88]. While mild symptoms and gastric emptying delays can often be treated with diet modification or symptom-directed medication alone, more severe symptoms and gastric emptying delays may require more advanced treatments and interventions. Nonetheless, treatment should generally be pursued in a stepwise fashion. The current and emerging treatments for gastric neuromuscular disorders are summarized in Table 2.

### 8.1. Glycemic Control

Treatment of the underlying cause is warranted if known or suspected, such as in diabetic gastroparesis, where tight glycemic control is recommended. While previously controversial, the Gastroparesis Clinical Research Consortium elucidated the benefit of tight glycemic control in diabetic gastroparesis, and standard management modalities such as continuous glucose monitoring and insulin pumps were safe and effective to employ [89]. Therefore, age-appropriate glycemic control should be pursued.

### 8.2. Nutrition

Dietary management is the foundation of treatment in mild-to-moderate cases, and nutritional support may be required in the most severe of cases. Malnutrition and electrolyte disturbances can be seen in uncontrolled disease or with flares. A small-particle-size low-fat diet is better tolerated by patients with gastroparesis and is recommended in major society guidelines [90]. Despite broad recommendations for dietary management, only around one third of patients received nutrition referrals after their gastroparesis diagnosis [91]. Partnership with a registered dietician to implement dietary changes can be instrumental in the success of patients with gastroparesis. Refractory patients with significant nutritional concerns may also require nutritional support via post-pyloric feeding such as endoscopic, radiological, or surgical gastrostomy with jejunal extension (G-J tube), but gastrostomy (G-tube) must be avoided as it can exacerbate the condition by delivering nutrition to a disordered stomach with delayed emptying. Lastly, exclusive total parenteral nutrition can be used in the most severe of cases, but use should be reserved as a last resort for patients refractory to all other means of nutrition and symptom management given the increased risks of infection and mortality [92].

### 8.3. Pharmacotherapy

Medications including promotility agents, neuromodulator agents, and antinausea agents can be used to address symptoms. Unfortunately, no medications exist to address the underlying pathophysiology and natural history of disease, though a great deal of research is underway in this pursuit. Regular electrocardiogram monitoring of the QTc interval is required with promotility, antinausea, and certain neuromodulator medications.

#### 8.3.1. Promotility Agents

Promotility agents accelerate gastric emptying, although the link between improvement in symptoms and gastric emptying remains unclear and may additionally be mediated through central antiemetic effects [2]. Metoclopramide, a dopamine D2 receptor antagonist and mixed serotonin 5-hydroxytryptamine (HT) 3 receptor antagonist and 5-HT4 receptor agonist, is a prokinetic and the only FDA-approved medication for gastroparesis, highlighting the need for more therapeutic options. Studies have revealed variable efficacy of metoclopramide ranging from no acceleration of gastric emptying to a 35% improvement in gastric emptying [93]. Though it carries a black box warning and is only approved for 12 weeks of use given the dreaded risk of tardive dyskinesia, experience has shown it can be employed continuously with frequent monitoring but should be used at the lowest effective dose [94]. Additionally, physicians and patients must be careful to avoid the use of concurrent dopamine receptor-acting agents given the rare risk of neuroleptic malignant syndrome. Erythromycin is a macrolide antibiotic and motilin receptor agonist that can be used at low doses as an effective prokinetic, with studies showing a 42.8 to 50% improvement in gastric emptying [93]. While erythromycin is 30–60% more effective in promoting gastric emptying than other promotility agents, its administration can inhibit P450 iso-enzymes, causing prolonged QT intervals associated with torsades de pointes and sudden cardiac death [93]. The medication is also prone to tachyphylaxis, which limits long-term use and may require “drug holidays”. Examples of possible schedules include 5 days of therapy followed by 2 days of drug cessation, or 2–3 weeks of therapy followed by 1 week of drug cessation.

#### 8.3.2. Neuromodulator Agents

Neuromodulators, though not free of side effects, may be better tolerated in the long term in comparison to prokinetics. Given the overlap or perhaps spectrum of disease with FD, neuromodulator agents should be considered and instituted early when appropriate. Important concepts in the use of neuromodulators are appropriate consideration of side effect profiles, adequate length of therapeutic trials, dose uptitration as tolerated, and medication stacking (using a combination of neuromodulator agents) to achieve therapeutic effects. Medications such as gabapentin and pregabalin, alpha-2 delta ligands that serve as antiepileptics and neuropathic pain modulators, are often utilized in pain-predominant patients because of their historical use in neuropathy, wide therapeutic index, and patient acceptability [95]. In a retrospective, open-label cohort study utilizing gabapentin for the treatment FD, improvement was noted in more than half of the subjects in overall, postprandial fullness, and upper abdominal pain subscores, and the subjects’ mean symptom scores decreased by almost half, with significant changes in all subscales (including upper abdominal pain, lower abdominal pain, postprandial fullness) except for bloating. However, 11.3% of patients discontinued gabapentin during the study, with 71.4% of the discontinuations due to side effects, most commonly dizziness and fatigue, showing that its use is not free of limitations.

In the setting of delayed gastric emptying, tricyclic antidepressants (TCAs) are not superior to other neuromodulators for the treatment of upper GI symptoms and may worsen frequently comorbid constipation [96]. Other traditional treatments for FD, such as selective serotonin reuptake inhibitors (SSRIs) or serotonin norepinephrine reuptake inhibitors (SNRIs), can be used in gastroparesis. Evidence suggests that paroxetine may selectively accelerate small intestinal transit, but it has also been associated with constipation [97]. Mirtazapine is a tetracyclic antidepressant that increases noradrenergic and serotonergic neurotransmission via blockade of central alpha-2 adrenergic receptors, resulting in an increased release of serotonin that stimulates serotonin 5-HT1 receptors in the setting of mirtazapine blocking 5-HT2 and 5-HT3 receptors, with possibly fewer neurotransmitter-related side effects than TCAs [98]. A prospective study of 30 patients showed that 76% of the patients reported improvements in nausea or vomiting symptoms; however, almost half of the patients experienced side effects, most commonly drowsiness and lethargy/fatigue, resulting in 20% of patients terminating treatment prior to completing the 4-week study period [98].

Buspar, a 5-HT1 receptor agonist, was only found to be helpful for symptoms of bloating in patients with gastroparesis, but it is thought to improve gastric accommodation and can be used more broadly for symptoms of FD [99]. Olanzapine, quetiapine, and aripiprazole are atypical antipsychotics with both D2 and 5-HT2A antagonist properties that have been used in functional GI disorders (FGIDs), particularly for nausea and abdominal pain symptoms, and are increasingly being used as augmentation in refractory gastroparesis [100]. A study of 21 patients receiving quetiapine for FGIDs refractory to an antidepressant alone found that nearly half of the patients discontinued the medication due to side effects or lack of effect; however, over half of the patients who continued the medication reported adequate symptom relief [101]. In one case report, aripiprazole reversed gastroparesis in a child with 1q21.1–q21.2 microdeletion [102]. However, due to additional histamine H1 and alpha-adrenergic receptor antagonism, as well as the anticholinergic properties of atypical antipsychotics, side effects including increased appetite, weight gain, cardiometabolic illness, and sedation can occur. Investigations into novel neurotransmitter receptor-acting therapies for GNDs are underway.

#### 8.3.3. Antinausea Agents

Antinausea medications that act on serotonin, dopamine, histamine, or neurokinin receptors might temporarily relieve symptoms, but do not address the underlying pathophysiology of the symptoms. Thus, they are most helpful in mild disease or in conjunction with other therapies. Ondansetron, a selective 5-HT3 receptor antagonist, was shown to reduce nausea from stomach distension without altering gastric compliance, volume, or accommodation in healthy volunteers, and has implications in pathways related to satiation [103]. However, ondansetron commonly causes constipation which can limit its use. Phenothiazide derivatives such as chlorpromazine, prochlorperazine, and promethazine can be used as antiemetics and exhibit action through dopamine D2 receptor antagonism, histamine H1 receptor antagonism, muscarinic M1 receptor antagonism, alpha-adrenergic receptor antagonism, and more [61]. However, they are prone to side effects of sedation, dry mouth, and constipation. Chlorpromazine and prochlorperazine are first-generation antipsychotics that are also efficacious in the treatment of nausea, primarily via central D2 receptor antagonism with no prokinetic activity, but carry a black box label cautioning increased mortality in elderly patients with dementia-related psychosis. Promethazine acts as an antiemetic through action as a histamine H1 receptor antagonist but is potentially habit-forming.

Aprepitant is a neurokinin-1 (NK-1) receptor antagonist that has been shown to act centrally to block receptor activation by substance P in the brain stem vomiting center, and is FDA-approved for chemotherapy-induced nausea and vomiting [104]. A randomized, double-blind, placebo-controlled study evaluating the effects of aprepitant on gastric motor function and satiation in healthy subjects found that it increases fasting, postprandial, and accommodation gastric volumes, as well as volume to fullness and maximum tolerated volume, suggesting that NK-1 receptors are implicated in the control of gastric volume and determining postprandial satiation and symptoms [105]. However, more long-term studies in patients with gastroparesis-like symptoms are needed. Scopolamine, a muscarinic M1 receptor antagonist that treats motion sickness and post-operative nausea and vomiting, has been used off-label for its antiemetic effect and has the benefit of transdermal administration, but has notable anticholinergic side effects such as dry mouth, vision changes, and drowsiness, and is not well studied in patients with GNDs [106]. While benzodiazepines like lorazepam have been shown to reduce nausea and vomiting, their use is limited by oversedation and dependence, limiting their use to monitored or inpatient settings [94].

### 8.4. Noninvasive Therapies

#### 8.4.1. Transcutaneous Auricular Vagus Nerve Stimulation (taVNS)

Transcutaneous auricular stimulation is a noninvasive electrical therapy that modulates Vagus nerve activity and has been shown to relieve gastroparesis symptoms with no side effects [107]. Currently, most patients must pay out of their pockets to obtain these particular devices, which limits their use. In rat models, invasive left cervical vagal nerve stimulation significantly accelerated gastric emptying by increasing pyloric relaxation and antral contraction amplitude [108]. A randomized subject-blinded, sham-controlled, cross-over study of 18 healthy subjects showed that active treatment with the gammaCore transcutaneous vagal nerve stimulator increased thresholds to bone pain, frequency of antral contractions, and gastroduodenal motility [109]. An open-label proof-of-concept study showed a 43% symptom response rate following 3–6 weeks of stimulation therapy (gammaCore, electroCore) in patients with severe drug-refractory gastroparesis [110]. These studies suggest taVNS as an emerging treatment option for GNDs and other gastrointestinal disorders with chronic pain or dysmotility. More studies are needed to determine if taVNS can predict response to gastric electrical stimulation, given their shared therapeutic pathways.

#### 8.4.2. Thoracic Spinal Magnetic Neuromodulation Therapy (ThorS-MagNT)

Repetitive magnetic stimulation of paraspinal regions has been shown to correct underlying bidirectional gut–brain neuropathy and sphincter dysfunction. A study of 33 patients with fecal incontinence demonstrated that translumbosacral neuromodulation was a safe and efficacious treatment with significant improvements in anorectal neuropathy, physiology, mechanics, and symptoms [111]. A proof-of-concept study aimed to improve the symptoms of gastroparesis through the reversal of sympathetic efferent spinogastric neuropathy and visceral afferent gastro-cortical pathways by targeting bilateral nerve roots at the mid-thoracic level with repetitive magnetic stimulation twice daily for five consecutive days [112]. All of the four patients treated at optimal intensities achieved a clinically significant reduction in total GCSI-DD score, with an average initial reduction of 74.9% that increased to an average reduction of 97.7% at 2 weeks post-treatment, most notably in symptoms of nausea, vomiting, and upper abdominal pain. While this treatment showed promising results in reducing symptom scores in a small number of patients with moderate–severe diabetic gastroparesis with no adverse events, it is not yet available outside of research centers.

### 8.5. Invasive Therapies

#### 8.5.1. Pyloric Botulinum Toxin (Botox) Injection

Pyloric Botox injection is not recommended in current society guidelines, as there is a lack of strong data to support its efficacy, with two small randomized control trials (RCTs) showing no difference between Botox administration and placebo [90,113]. Furthermore, it is not a durable treatment option, as the theoretical response lasts only a few months, and repeated injections can cause fibrosis, which presents challenges if pyloric myotomy is later pursued. However, given the limited treatment options in refractory gastroparesis, some providers and institutions still administer pyloric Botox due to prior promising studies, including a retrospective review of 52 patients who underwent pyloric botulinum toxin injection, that showed a 43% symptomatic response rate that endured a mean of 5 months [114]. Despite some data which suggested that a clinical response to Botox may predict a response to gastric per oral pyloromyotomy (G-POP), it is not recommended to select candidates for G-POP based on an intervention that lacks thorough supportive evidence, and studies are underway to illuminate more viable predictors of treatment response [115].

#### 8.5.2. Gastric per Oral Pyloromyotomy (G-POP)

More invasive treatments are reserved for patients with more refractory disease and significant impairments in quality of life. G-POP, also known as gastric per oral endoscopic myotomy (G-POEM), has been well validated in improving gastric emptying and providing symptom relief in refractory gastroparesis, particularly in the diabetic subgroup [116]. Nausea and vomiting predominant patients benefited most from G-POP, while pain-predominant patients experienced limited symptomatic success, and pain often persisted. Pain symptoms may therefore be more related to FD and respond more satisfactorily to FD-focused treatments. While studies demonstrate that elevated baseline GCSI scores and gastric retention >20% at 4 h were associated with improved clinical response to G-POP, more studies are needed to define which populations will respond and benefit most from G-POP [117]. Although there is little difference in the symptoms, physiology, and treatment of gastroparesis and FD as GNDs, G-POP would only be considered in a patient with a documented delay in gastric emptying with refractory symptoms. A systematic review and meta-analysis showed that G-POP was a similarly efficacious but more cost-effective approach than surgical pyloromyotomy in managing refractory gastroparesis [118]. Emerging evidence supports the use of G-POP over surgical myotomy in lung transplant patients with refractory gastroparesis [119]. When patients fail to respond to G-POP, repeat G-POP or surgical pyloromyotomy are often considered.

#### 8.5.3. Gastric Electrical Stimulation (GES)

Gastric electrical stimulation involves a device implanted into the abdomen with leads stimulating the stomach and pylorus and is subsequently more invasive and eventually requires battery replacement. The full mechanism of action is unclear and may involve electrical neuromodulation of a variety of pathways including visceral hypersensitivity (efferent nociceptive signaling) and gastric (electrical) dysrhythmia. While sixteen open-label studies have shown an improvement in total symptom score with GES, five studies with random allocation to periods with or without GES failed to show a change in symptom scores between these periods [120]. Though the Entera GES system was shown to be generally safe and improve gastric emptying, their placement and use are not free from complications such as skin erosion, wound dehiscence, or device migration/flipping [121]. A temporary endoscopic gastric electrical stimulator can be trialed prior to permanent stimulator placement. The Gastroparesis Clinical Research Consortium demonstrated no meaningful symptomatic improvement over time, except with regard to nausea (improved by ≥1 point, RR-1.31; *p* = 0.04) [122]. Gastric electrical stimulation currently only holds a low conditional recommendation as a compassionate humanitarian-use device, with greatest efficacy observed in diabetic gastroparesis [90]. The combination of gastric electrical stimulator placement with pyloromyotomy may be beneficial in some groups, particularly in the diabetic gastroparesis group [123].

#### 8.5.4. Surgical Intervention

Surgical pyloroplasty and even partial gastrectomy have historically been performed for severe refractory cases of gastroparesis, but they have largely been replaced by aggressive medical therapy and G-POP, which is noninferior to surgical pyloromyotomy. Roux-en-Y gastric bypass (RYGB) has been performed in select patients with both refractory gastroparesis and obesity, with 71% reporting improved symptoms at the last follow-up [121]. A systematic review found favorable results of both sleeve gastrectomy and RYGB for improving symptoms (particularly nausea and vomiting) and gastric emptying in refractory gastroparesis [124]. The results of these studies must be interpreted in the context of only a small number of the most severe and refractory of patients undergoing these invasive interventions. However, this reinforces the notion that gastroparesis is a debilitating condition that significantly impacts quality of life and drives patients to pursue invasive or even experimental therapies in hopes of alleviating their symptoms. Furthermore, surgical intervention is not without risks including duodenal stump leak and more, although no 30-day mortality was reported in one review. While surgical intervention may still play a role in the management of the most refractory cases of gastroparesis including surgical pyloromyotomy after G-POP failure, it is rarely performed and only in carefully chosen patients.

### 8.6. Complementary Therepies

#### 8.6.1. Gut–Brain Axis Modulation

Additionally, the role of the gut–brain axis cannot be overstated when discussing the overlap of gastroparesis and functional dyspepsia as GNDs. Therapies that modulate neural pathway signaling can be integral in preserving quality of life, including cognitive behavioral therapy, hypnosis, acupuncture, meditation, tai chi, yoga, and more. Cognitive behavioral therapy has shown symptomatic benefits when used alone or in combination with standard FD treatment, although more studies are needed to delineate the role of CBT in patients with gastroparesis [125]. CBT is also relevant for ARFID, which can be associated or overlap with GNDs [126]. Acupuncture and electroacupuncture have been shown to affect the GI system through complex neuroimmune–endocrine mechanisms, including GI motility, inflammation, visceral hypersensitivity, GI barrier function, and microbiota [127]. In a study of 35 patients with diabetic gastroparesis, acupuncture was shown to provide symptom relief in 94% of the patients, which was significantly higher than the control groups receiving domperidone or nothing [128]. A large review of gut-directed hypnosis, typically administered in 6 to 12 weekly sessions, found it was a highly effective and adaptable treatment option in refractory FGIDs, but is limited by time requirements, cost, general availability, and lower effectiveness when delivered outside specialized research centers [129,130]. Emerging options are becoming available to increase access to gut–brain axis modulation, such as internet-delivered CBT or app-delivered gut hypnosis. Gut–brain axis modulation has been underutilized in the treatment of GNDs, and whichever modalities are most acceptable and accessible to patients can be pursued in addition to standard medical treatment.

#### 8.6.2. Herbal Formulations

Herbal remedies such as Iberogast have been shown to improve FD symptoms compared to placebo, and can be trialed in patients with gastroparesis [131]. Studies in obesity-induced diabetic mice support the role of curcumin (the active ingredient in turmeric) in modulating nitrergic-mediated gastric motility and gastric emptying via normalizing inflammation and oxidative stress through activation of nuclear factor erythroid 2-related factor 2, which increases neuronal NOS gene expression and function [132]. OLNP-06 is a concentrated ginger extract formulation that was shown to improve dyspepsia symptoms (such as postprandial fullness, upper abdominal bloating, and early satiation) in 79% of patients, with 64% of the patients reporting complete elimination of symptoms in a 4-week randomized, double-blind, placebo-controlled study [133]. Given the promising results with little to no adverse events, further studies are needed to explore the role of nutraceuticals in GNDs.

#### 8.6.3. Cannabinoids

The role of Cannabis or cannabinoid products in the management of gastroparesis is unclear. While studies show it can delay gastric emptying, many patients have reported significant relief of their GI symptoms [134]. Dronabinol, synthetic delta-9-tetrahydrocannabinol (THC), is FDA-approved for chemotherapy-induced nausea and vomiting; though its central actions are only partially understood, it stands to reason that it may be efficacious for other causes of nausea and vomiting as well. THC occurs naturally in Cannabis sativa or marijuana plants and activates endogenous CB1 and CB2 receptors, with the majority of the psychoactive effects (affective, sensory, somatic, cognitive alterations) and therapeutic benefits (analgesia, appetite enhancement, muscle relaxation, hormonal actions) mediated by the CB1 receptor, and the majority of the immunomodulatory properties mediated by the CB2 receptor [135]. Cannabidiol (CBD) is also found naturally in Cannabis and acts as a partial CB2 receptor agonist and low-affinity CB1 antagonist, with central effects that can reduce gut sensitivity and inflammation, and does not face the same legal restrictions as products containing THC. An RCT showed the efficacy of CBD in reducing GCSI (*p* = 0.008), the number of daily vomiting episodes (*p* = 0.006), inability to finish a normal-sized meal (*p* = 0.029), and overall symptom severity (*p* = 0.034) [136]. The role of cannabinoid receptor modulation in the treatment of GNDs requires more structured studies before conclusions can be drawn. Regardless, in light of broad medical legalization and rising recreational legalization in the United States, the number of patients using cannabis products may continue to rise, and local considerations can impact its safety and reliability as a therapeutic option in GNDs [137].

### 8.7. Treatments to Avoid

Treatments that should be avoided in gastroparesis include opioids or other medications that can slow GI transit, as the short-term benefit in pain management is far outweighed by the ultimate effects of worsening gastric emptying and exposing the patient to the unnecessary risks of opioids, including dependence, cardiopulmonary depression, and death. Notably, opioid use was associated with increased gastric retention, worse quality of life, increased hospitalization, and increased antiemetic and pain modulator medication use compared to nonusers [138]. In general, escalation to more invasive therapies without prolonged trials of neuromodulators including stacking and uptitration should be avoided given the risk of overproceduralization without a substantial benefit in symptoms or quality of life. As previously stated, G-tube placement is contraindicated as it does not bypass the disordered stomach and may exacerbate symptoms. Additionally, the finding of celiac artery compression or median arcuate ligament syndrome is typically an incidental diagnosis and not a cause of symptoms, and pursuing aggressive investigations and treatment may not address symptoms.

### 8.8. Novel and Investigational Therapies

Newer agents are under investigation that target diverse gastric (fundic, antral, and pyloric) motor functions, including novel serotonergic 5-HT4 agonists, dopaminergic D2/3 antagonists, neurokinin NK-1 antagonists, and motilin and ghrelin agonists. Since HO1 expression is protective against ICC loss, it has been theorized that macrophage HO1 induction could be a therapeutic target in reversing ICC loss and delayed gastric emptying [46,139]. In patients with suspected immune dysfunction or dysregulation, corroborative testing and a trial of immunotherapy may improve symptoms. Additionally, there may be a role for experimental interventions in the most refractory cases, such as celiac plexus block for chronic abdominal pain severely affecting quality of life and nutrition.

#### 8.8.1. Emerging Promotility Agents

Domperidone, another D2 receptor antagonist, is effective in improving nausea, vomiting, early satiety, and gastric emptying, with fewer central side effects than metoclopramide [140]. However, it is not FDA-approved due to risks of QT prolongation and sudden death; it can only be officially obtained by patients in the US under the Expanded Access to Investigational Drugs program through the FDA. Anecdotally, domperidone has been obtained by patients through international pharmacies, but caution must be exercised given the risk of unknown contents. Deuterated domperidone may decrease toxicities and side effects and is currently being investigated in clinical trials. Prucalopride, a selective 5-HT4 receptor agonist, is used off-label for gastroparesis, as it is only approved for chronic idiopathic constipation (CIC), despite its endorsement by society guidelines; however, many patients with gastroparesis also experience CIC and may be candidates for prucalopride [141].

Azithromycin, another macrolide antibiotic with motilin receptor activity, is emerging as an efficacious alternative to erythromycin with equivalent profiles for accelerating gastric emptying in patients with gastroparesis, with the added benefit of a longer duration, a better side effect profile, and a lack of P450 interaction and the associated risks [142]. Given these potential benefits and its wide availability, some providers already employ azithromycin off-label for gastroparesis.

HM01, an orally active ghrelin receptor agonist that crosses the blood–brain barrier, has been shown to activate central vagal and myenteric pathways in rat models with abdominal surgery-induced gastric inflammation and ileus [143]. A systematic review and meta-analysis evaluating the efficacy and safety of ghrelin agonists in diabetic gastroparesis found that ghrelin administration significantly improved overall gastroparesis symptoms, nausea, vomiting, early satiety, and abdominal pain compared to placebo, with no difference in adverse events [144]. However, more studies are needed before ghrelin agents become available for use.

#### 8.8.2. Immunomodulator Agents

One study proposed neural autoimmunity as an under-recognized etiology of gastroparesis and trialed immunotherapy to aid in symptom relief and diagnosis in patients refractory to medications and stimulators, with suspected autoimmune gastrointestinal dysmotility (based on serological evidence or personal/family history of autoimmune disease) [49]. The study demonstrated that 74% of the patients achieved symptomatic and/or scintigraphic improvement post-immunotherapy (6–12 weeks of intravenous immune globulin (IVIg), or methylprednisolone, or both). A subsequent study in a more targeted population of patients refractory to medications and stimulators, with suggested neuroinflammation on full-thickness gastric biopsy and glutamic acid decarboxylase (GAD) 65 positive autoantibodies, showed a maximal response to IVIg (67%), with more than half (55%) of the patients noting improvement in vomiting and almost half (45%) noting improvement in nausea, abdominal pain, and bloating [44]. Pyridostigmine, an acetylcholinesterase inhibitor, was also found to reduce symptoms in a patient with gastroparesis due to underlying autoimmune etiology [7]. These combined findings support the use of immunotherapy in patients refractory to available treatments with suspected autoimmune contributions to their disease.

#### 8.8.3. Celiac Plexus Block (CPB)

Celiac plexus block can be performed with endoscopic ultrasound or noninvasive fluoroscopic or ultrasound guidance, and has been shown to improve pain and increase GI motility by modulating visceral afferent fibers that innervate from the distal esophagus to the transverse colon [145,146]. CPB has been shown to improve symptoms of chronic abdominal pain in case reports of patients with pain-predominant severe refractory gastroparesis, with additional benefits of increased feeding tolerance and avoiding or eliminating opioid use. While one case report showed a significant reduction in pain score following the administration of a thoracic splanchnic nerve block followed by radiofrequency thermocoagulation at the T11 and T12 vertebral levels to target parasympathetic neuropathy in a patient with FD, there are no published data for its use in gastroparesis [147].

## 9. Conclusions

Gastroparesis is a complex disorder with diverse etiologies, significant symptom burden, and limited treatment options. Gastroparesis can be a debilitating condition for patients to live with, and recognizing the concurrent role of FD can greatly impact the treatment options provided to patients and thus their quality of life. Advances in understanding the shared pathophysiology and management of GNDs are promising but underscore the need for individualized, multidisciplinary care approaches. Multiple options exist to diagnose delayed gastric emptying, most commonly gastric scintigraphy followed by breath testing; whichever is most feasible (based on local availability and insurance coverage) can be pursued. Promising new modalities exist to diagnose gastroparesis and whole-gut motility disturbances, such as electrogastrography and wireless motility capsules, which may become more widely used provided they are covered by insurance and FDA-cleared, respectively.

Dietary modifications, such as small-particle-size and low-fat diets, can be implemented by all patients with the support of a dietician if accessible. Prokinetics and antiemetics provide benefit to some patients, but neuromodulators are historically underused in gastroparesis. Given the overlap or perhaps spectrum of disease between gastroparesis and FD as GNDs, neuromodulation should be more broadly considered in their management. Noninvasive treatments, such as auricular nerve stimulation and experimental thoracic magnetic neurostimulation, show promising results as adjunctive therapies in restoring normal motility and alleviating symptoms. Advanced therapies such as G-POP and gastric electrical stimulation can be effective but should be reserved for patients with refractory disease severely impairing quality of life. Post-jejunal nutrition, exclusive parenteral nutrition, and surgical intervention remain options for the most refractory of patients with significant nutritional concerns. Figure 2 summarizes the proposed diagnostic and treatment algorithm for patients with GNDs.

There is an urgent need for more targeted, efficacious, and durable treatment options in gastroparesis. More research is also needed to elucidate the causative pathways of disease, find reliable biomarkers, understand sex differences, and improve long-term outcomes. Novel biomarkers could enable earlier diagnosis and identification of distinct disease subtypes, guiding targeted therapies based on specific pathophysiological mechanisms. Genetic profiling may help stratify patients into subgroups with distinct therapeutic responses, allowing for precision medicine strategies tailored to individual genetic susceptibilities. As these innovations progress, an integrated, patient-specific approach could optimize outcomes and improve quality of life. Ultimately, gastroparesis and FD are complex GNDs with significant overlap in pathophysiology, and thus, treatment requires nuanced care from providers to achieve successful treatment outcomes.

## Figures and Tables

**Figure 1 diagnostics-15-00935-f001:**
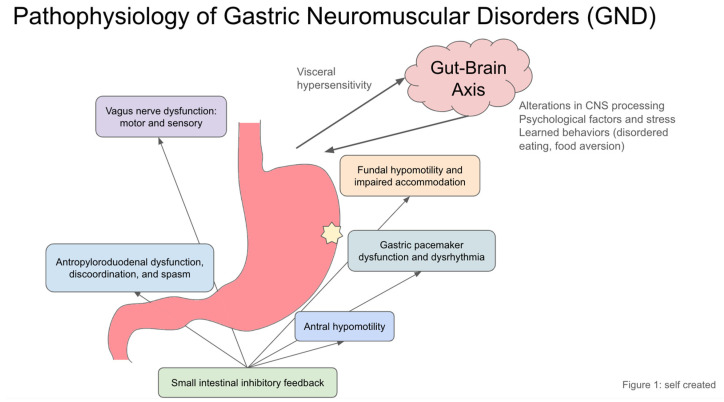
Pathophysiology of gastric neuromuscular disorders.

**Figure 2 diagnostics-15-00935-f002:**
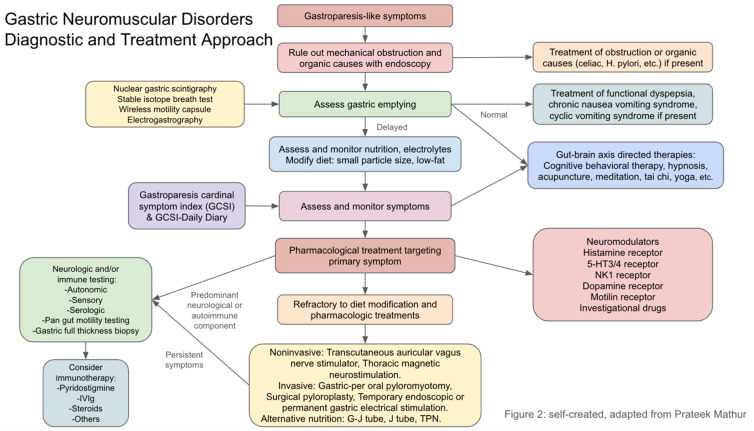
Gastric neuromuscular disorders diagnostic and treatment approach [148].

**Table 1 diagnostics-15-00935-t001:** Current and emerging diagnostics for gastric neuromuscular disorders.

	Diagnostic Information	Clinical Utility	Additional Information
**Nuclear gastric scintigraphy**	Gastric emptying May provide data on gastric accommodation	Most widely available 4-h testing and food intolerances limit use	Gold standard in assessment of gastric emptying
**Gastric emptying breath Test**	Gastric emptying	Moderately widely available 4-h testing	May be affected by physical activity, malabsorption, and advanced lung, liver, or cardiac disease
**Wireless motility capsule**	Gastric emptying Whole gut motility data	Noninvasive ambulatory testing Atmo Capsule pending FDA clearance	Avoid with gastrointestinal stenosis or stricture Caution with post-surgical anatomy
**Electrogastrography**	Gastric emptying Characteristic symptom phenotype data	Noninvasive ambulatory testing Gastric Alimetry and GutTracker are FDA cleared	Out of pocket cost Provide data regarding small bowel and colon activity
**Antroduodenal manometry**	Assessment of strength and coordination of muscle contractions Characteristic patterns among gastroparesis etiologies: idiopathic, diabetic, and post-surgical	Typically performed in research settings only Limited use given invasive nature and etiologies often determined by empiric means	Not diagnostic of gastroparesis Pylorospasm can be seen in diabetic gastroparesis and recurrent nausea and vomiting Can diagnose small intestinal pseudo-obstruction
**Pyloric EndoFLIP**	Low pyloric compliance and Cross-Sectional Area seen in gastroparesis	Limited use given inconclusive findings Low pyloric distensibility may predict response to G-POP	Not diagnostic of gastroparesis
**Liquid satiety testing**	Functional assessment of gastric accommodation and sensation	Typically performed in research settings only Limited use given impaired accommodation and sensation often determined by empiric means	Not diagnostic of gastroparesis
**Gastric MRI**	Functional gastric imaging including neurogastric signaling	Available in research settings only	Not diagnostic of gastroparesis

**Table 2 diagnostics-15-00935-t002:** Current and emerging treatments for gastric neuromuscular disorders.

	Clinical Role	Treatment Positioning and Patient Selection	Additional Considerations
**Glycemic control**	Manages underlying hyperglycemia contributing to nerve dysfunction	Foundational Indicated for all patients with elevated A1c	Tight glycemic control, continuous glucose monitoring, and insulin pumps are safe and effective to employ when indicated
**Nutrition**	Small particle size, low-fat diet improves oral tolerance	Foundational Indicated for all patients including dietician referral	G-J tube or TPN only in the most refractory of cases with significant weight loss, electrolyte disturbances
**Neuromodulator agents**	Improve gastroparesis-like symptoms Modulate nerve dysfunction, disordered gut-brain interaction	First line Can trial in all patients targeting predominant symptom	Appropriate side effect profiles, adequate length of therapeutic trials, dose uptitration as tolerated, and medication stacking if needed
**Promotility agents**	Improve gastric emptying and gastroparesis-like symptoms	First line Can trial in all patients	Monitor QTc and tardive dyskinesia New agents may become available
**Antinausea agents**	Improve nausea and vomiting	First line Can trial in patients with predominant nausea vomiting	Monitor QTc New agents may become available
**Gut-brain axis modulation**	Improves gastroparesis-like symptoms Alters neural pathway signaling in disordered gut-brain interaction	Complementary Can trial in all patients	Trial based on patient acceptability and accessibility Cognitive behavioral therapy, gut hypnosis, acupuncture, etc.
**Herbal formulations**	Improve dyspepsia symptoms May modulate gastric emptying	Complementary Can trial in all patients	Trial based on patient acceptability and accessibility
**Cannabinoids**	Improve nausea, vomiting, and inability to finish a meal	Complementary Can trial in patients with medical marijuana or legal age	Trial based on local availability, patient acceptability and accessibility Sublingual or ingested agents may be preferred
**Immunomodulator agents**	Improve nausea, vomiting, abdominal pain, bloating	Can trial in patients with known or suspected autoimmune gastrointestinal disorders refractory to other treatments	Consider neurologic or immune testing such as autonomic, sensory, serologic, pan gut motility, or gastric full thickness biopsy
**Transcutaneous auricular Vagus nerve stimulation**	Improves antral contractions, gastroduodenal motility, pain thresholds, and gastroparesis-like symptoms	Can trial in all patients refractory to other treatments	Out of pocket cost Noninvasive with short term effect that requires repeated or daily use
**Thoracic spinal magnetic neuromodulation therapy**	Improves symptoms particularly nausea, vomiting, and upper abdominal pain	Available in research settings only for severe refractory gastroparesis	Noninvasive with prolonged effect but requires repetitive stimulation over days
**Pyloric botulinum toxin (Botox) Injection**	May improve gastroparesis-like symptoms, pyloric dysfunction, and gastric emptying	Not guideline recommended Some centers still use in severe refractory gastroparesis	Prolonged effect can last for months but requires repeated endoscopy May cause fibrosis
**Gastric per oral pyloromyotomy (G-POP)**	Improves gastric emptying and symptoms particularly nausea and vomiting	Consider in severe predominant nausea vomiting gastroparesis refractory to other treatments with impaired quality of life (QOL)	Only consider with documented delay in gastric emptying Noninferior to surgical pyloromyotomy and less invasive therefore preferred, particularly for lung transplant patients with refractory gastroparesis
**Gastric electrical stimulation**	Improves nausea	Consider in severe gastroparesis refractory to other treatments with impaired QOL	Low conditional guideline recommendation as humanitarian use device most efficacious in diabetic gastroparesis
**Celiac plexus block**	Improves pain and increases gastrointestinal motility	Not guideline recommended	Consider with caution in pain predominant patients refractory to other treatments with impaired QOL
**Surgical intervention**	Improves gastric emptying and symptoms particularly nausea and vomiting	Not guideline recommended	Consider surgical pyloromyotomy in predominant nausea vomiting gastroparesis refractory to other treatments with impaired QOL if G-POP is not available, G-POP was failed, or there is an additional indication or preference for surgery

## Data Availability

No new data were created or analyzed in this study. Data sharing is not applicable to this article.

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
