# Peer review of "Current Concepts in Gastroparesis and Gastric Neuromuscular Disorders—Pathophysiology, Diagnosis, and Management"

_diagnostics, 2025, doi:10.3390/diagnostics15070935_

Round 1
Reviewer 1 Report
Comments and Suggestions for Authors
This manuscript provides a comprehensive review of gastroparesis and gastric neuromuscular disorders, covering their pathophysiology, diagnosis, and management. It highlights the overlap between gastroparesis and functional dyspepsia, emphasizing the need for a broader approach to classification and treatment. The paper discusses various diagnostic tools, including gastric emptying tests and emerging technologies, while also addressing the limitations of current methods. Treatment strategies range from dietary modifications and pharmacotherapy to invasive procedures like gastric per-oral pyloromyotomy (G-POP) and gastric electrical stimulation. Overall, the manuscript underscores the complexity of these disorders and the ongoing need for more effective diagnostic and therapeutic approaches. Here are some minor comments that should be addressed.
- In the abstract, the author may clearly state the objective, main findings, and implications.
- In lines 28–48, the overlap between gastroparesis and functional dyspepsia is well explained. However, why is distinguishing them still important? A brief statement on the clinical relevance would be more reasonable.
- In lines 49–67, this section includes useful prevalence data. However, some statistics lack citations. Adding references would make the statement more valid, especially for the differences between type 1 and type 2 diabetes-related gastroparesis.
- In the lines 118–133, the discussion about gastric emptying and symptom severity is important. However, what does this mean for clinical practice? Should gastric emptying tests be used less for diagnosis? The author may add a short clarification.
- In the lines 406–427, this section highlights limitations, are there newer methods that offer better accuracy? If so, the authors may add brief sentences to describe it.
Author Response
Comment 1: In the abstract, the author may clearly state the objective, main findings, and implications.
Response 1: We agree. Your point is well taken and the text has been updated accordingly. Lines 16-17:
”This article aims to review the current concepts in gastroparesis and GNDs including pathophysiology, diagnosis, and management.”
Comment 2: In lines 28–48, the overlap between gastroparesis and functional dyspepsia is well explained. However, why is distinguishing them still important? A brief statement on the clinical relevance would be more reasonable.
Response 2: We agree and appreciate this point. The text has been revised to emphasize this point. Lines 44-46:
“Unless severe and persistent, delayed gastric emptying may represent a marker of underlying gastric neuromuscular dysfunction rather than the cause of symptoms.”
Comment 3: In lines 49–67, this section includes useful prevalence data. However, some statistics lack citations. Adding references would make the statement more valid, especially for the differences between type 1 and type 2 diabetes-related gastroparesis.
Response 3: Agreed, thanks for noting. This point is critical and references have been updated as needed. Lines 61-71:
“Women and Caucasians were noted to have the highest prevalence of gastroparesis, and multiple studies support the female predominance of gastroparesis with as high as 80% of cases affecting female individuals[5,7,8]. Diabetes mellitus accounts for a significant proportion of cases, particularly type 1 diabetes, followed by idiopathic and post-surgical causes. Studies report that up to 12% of patients with diabetes experience symptoms consistent with gastroparesis, but not all undergo confirmatory testing[9]. The overall prevalence of gastroparesis in the US population is estimated at 0.16%, with notable in-creases in prevalence associated with type 1 diabetes mellitus (4.59%) and type 2 diabetes mellitus (1.31%)[6]. However, given the large number of patients with type 2 diabetes in the US, the number of patients in that group may range from 3- to 9-fold greater than the type 1 diabetes group[6,10].”
Comment 4: In the lines 118–133, the discussion about gastric emptying and symptom severity is important. However, what does this mean for clinical practice? Should gastric emptying tests be used less for diagnosis? The author may add a short clarification.
Response 4: We agree with this point and are grateful for your input. We added a clarification as suggested. Lines 138-144:
“While severe and persistent gastric emptying delays may contribute to symptomatology, mild and/or intermittent delays may serve clinically as an indicator of underlying gastric neuromuscular dysfunction as opposed to the cause of symptoms. Therefore, gastric emptying may be reassessed over time, most reasonably if there are significant changes in symptoms, generally no sooner than at 6 months, and more judiciously following multiple years.”
Comment 5: In the lines 406–427, this section highlights limitations, are there newer methods that offer better accuracy? If so, the authors may add brief sentences to describe it.
Response 5: Thank you for this point, we agree and the text has been revised to this point. Lines 430-432:
“While limitations exist, nuclear gastric scintigraphy remains the gold standard in assessing gastric emptying and the accuracy of newer testing modalities is validated against it.”
Reviewer 2 Report
Comments and Suggestions for Authors
Gastroparesis is a topical issue in modern gastroenterology. At the same time, information on pathogenesis is not entirely clear and approaches to therapy have not been clearly developed. Taking this into account, this review article seems very relevant and practically significant. The work is methodologically competent, well structured and will be of interest to a wide range of readers. At the same time, in order to improve the quality of the manuscript, I would like to make several suggestions to the authors.
In the etiology section, update the data and add more recent sources of literature. I would like to see the ratio of etiological factors in the structure of gastroparesis incidence - idiopathic, diabetic, etc.
In my opinion, it is advisable to expand the differential diagnosis of clinical symptoms, including in the age aspect. In the drug treatment (prokinetics) section, clarify the effectiveness of therapy with the level of evidence of data, provide links to systematic reviews and meta-analyses.
Author Response
Comment 1: In the etiology section, update the data and add more recent sources of literature. I would like to see the ratio of etiological factors in the structure of gastroparesis incidence - idiopathic, diabetic, etc.
Response 1: We agree and appreciate this point. The text has been revised to address this point. Lines 93-95:
“Gastroparesis can be classified into idiopathic (11.3% - 49.4%), diabetic (25.3% - 78.1%), post-surgical (15.0%), and less commonly, medication-induced or viral sub-types[8,10].”
Comment 2: In my opinion, it is advisable to expand the differential diagnosis of clinical symptoms, including in the age aspect.
Response 2: Thank you for pointing this out. We agree and have updated the text accordingly. Lines 580-586:
“Obtaining a comprehensive history and physical is paramount in the diagnosis of GNDs and helps guide the evaluation which usually includes endoscopy and imaging. Age and gender appropriate investigations must also be undertaken. For example, providers may recommend pelvic ultrasound for a woman of any age with early satiety, bloating, and abdominal pain to exclude an ovarian lesion in the appropriate clinical setting. After the exclusion of structural and organic causes of symptoms, additional disorders of gut-brain interaction can be considered.”
Comment 3: In the drug treatment (prokinetics) section, clarify the effectiveness of therapy with the level of evidence of data, provide links to systematic reviews and meta-analyses.
Response 3: We agree with this point and are grateful for your input. We added clarification as recommended. Lines 651-653, 656-658, and 664-665:
“Promotility agents accelerate gastric emptying, although the link between im-provement of symptoms and gastric emptying remains unclear and may additionally be mediated through central antiemetic effects[2].”
“Studies have revealed variable efficacy of metoclopramide ranging from no acceleration of gastric emptying to 35% improvement in gastric emptying[92].”
“Erythromycin is a macrolide antibiotic and motilin receptor agonist that can be used at low doses as an effective prokinetic, with studies showing 42.8 to 50% improvement in gastric emptying[92].”
Reviewer 3 Report
Comments and Suggestions for Authors
Well written review regarding an important topic that is at times challenging to navigate and treat in the clinical setting. Although not novel it is extremely relevant to practice. The topic is clear and discussed in detail and easy to follow. It provides a good overview and is very relevant to clinical practice for the gastroenterologist as well as the primary care provider. The manuscript is supported by use of reputable references. Lastly, they have provided good illustrations as well.
Author Response
Comment 1: Well written review regarding an important topic that is at times challenging to navigate and treat in the clinical setting. Although not novel it is extremely relevant to practice. The topic is clear and discussed in detail and easy to follow. It provides a good overview and is very relevant to clinical practice for the gastroenterologist as well as the primary care provider. The manuscript is supported by use of reputable references. Lastly, they have provided good illustrations as well.
Response 1: We appreciate and thank you for your review of our manuscript and included figures. No revisions have been made in response to this comment.